# Development and application of a high-content virion display human GPCR array

Guan-Da Syu [1,2,3], Shih-Chin Wang[4], Guangzhong Ma [5], Shuang Liu[1,2], Donna Pearce[6], Atish Prakash[6], Brandon Henson[3], Lien-Chun Weng[3], Devlina Ghosh[7], Pedro Ramos[8], Daniel Eichinger[8], Ignacio Pino [8], Xinzhong Dong [7,9], Jie Xiao[4], Shaopeng Wang [5], Nongjian Tao[5,10], Kwang Sik Kim[6], Prashant J. Desai[3] & Heng Zhu [1,2,3]

Human G protein-coupled receptors (GPCRs) respond to various ligands and stimuli. However, GPCRs rely on membrane for proper folding, making their biochemical properties difficult to study. By displaying GPCRs in viral envelopes, we fabricated a Virion Display (VirD) array containing 315 non-olfactory human GPCRs for functional characterization. Using this array, we found that 10 of 20 anti-GPCR mAbs were ultra-specific. We further demonstrated that those failed in the mAb assays could recognize their canonical ligands, suggesting proper folding. Next, using two peptide ligands on the VirD-GPCR array, we identified expected interactions and novel interactions. Finally, we screened the array with group B *Streptococcus*, a major cause of neonatal meningitis, and demonstrated that inhibition of a newly identified target, CysLTR1, reduced bacterial penetration both in vitro and in vivo. We believe that the VirD-GPCR array holds great potential for high-throughput screening for small molecule drugs, affinity reagents, and ligand deorphanization.

[1] Department of Pharmacology and Molecular Sciences, Johns Hopkins University School of Medicine, Baltimore, MD 21205, USA. [2] Center for High-Throughput Biology, Johns Hopkins University School of Medicine, Baltimore, MD 21205, USA. [3] Viral Oncology Program, Department of Oncology, The Sidney Kimmel Comprehensive Cancer Center, Johns Hopkins University School of Medicine, Baltimore, MD 21231, USA. [4] Department of Biophysics and Biophysical Chemistry, Johns Hopkins University School of Medicine, Baltimore, MD 21205, USA. [5] Biodesign Center for Bioelectronics and Biosensors, Arizona State University, Tempe, AZ 85287, USA. [6] Division of Paediatric Infectious Diseases, Johns Hopkins University School of Medicine, Baltimore, MD 21287, USA. [7] Solomon H. Snyder Department of Neuroscience, Johns Hopkins University School of Medicine, Baltimore, MD, USA. [8] CDI Laboratories, Inc., Mayaguez, Puerto Rico 00682, USA. [9] Howard Hughes Medical Institute, Johns Hopkins University School of Medicine, Baltimore, MD 21205, USA. [10] School of Electrical, Computer and Energy Engineering, Arizona State University, Tempe, AZ 85287, USA. Correspondence and requests for materials should be addressed to K.S.K. (email: kwangkim@jhmi.edu) or to P.J.D. (email: pdesai@jhmi.edu) or to H.Z. (email: hzhu4@jhmi.edu)

The G protein-coupled receptor (GPCR) family is comprised of >800 members, forming the largest membrane protein family in humans. Almost all of the annotated GPCRs function via binding to agonists or antagonists, ranging from ions to metabolites, small molecules, peptides and proteins, and exhibit a wide variety of signaling pathways. As archived in the IUPHAR/BPS database, GPCRs are classified into three large classes (i.e., Classes A, B, and C) and two smaller classes (i.e., Frizzled and Adhesion). Of all annotated human GPCRs, 370 are non-odorant and play crucial roles in many different aspects of cellular processes. They are the most preferred drug targets because their dysregulation often lead to disease or cancer. In fact, of the ~1400 FDA-approved drugs, 475 (34%) target 108 non-odorant GPCRs[1–3]. Although 70% of them have known ligands, a significant fraction of them are orphans without an identified ligand[1–3].

However, non-odorant GPCRs are notoriously difficult to study for several reasons. First, GPCRs must be embedded in a membrane to maintain proper conformation, and many also require a series of post-translational modifications, such as glycosylation, for activity[4]. Second, despite the recent progress in the cloning and expression of GPCR proteins in cells, their expression levels vary greatly, hampering an effective strategy to study these proteins in parallel and on a genomic scale[5]. Third, a considerable number of GPCRs are toxic to the host cells once overexpressed, making it difficult to establish stable cell lines that constitutively express these receptors[6]. Finally and perhaps most importantly, 116 (31%) of the ~370 non-odorant GPCRs are orphans, without known physiological agonists or antagonists, which precludes study of their activity.

Because of the significant importance of these druggable GPCRs, it is critical to develop scalable technology platforms to enable their characterizations in a high-throughput fashion. Human protein arrays carrying GPCRs are available from several commercial sources, such as Invitrogen (Life Sciences), Origen Inc., and CDI Laboratories. However, GPCRs spotted on these commercial arrays are unlikely to maintain their native conformation or activities because detergents were used to purify these proteins. In a series of publications, Fang et al.[7,8] at Corning Inc. described fabrication of human GPCR arrays by spotting down membrane preps of mammalian cells in which a particular GPCR-of-interest was overexpressed. However, membrane preparations used to fabricate the GPCR arrays are unavoidably contaminated with other endogenously expressed GPCRs, raising a homogeneity issue. Moreover, it is almost impossible to control concentration or orientation of the GPCRs on the array surface. Finally, because of stability issues caused by spotting fractured membrane preps, oligosaccharides were often added to associate with the head groups of the lipid bilayers, resulting in a higher susceptibility to buffer composition and pH.

As a proof-of-principle, we developed Virion Display (VirD) technology with which both single (i.e., CD4) and multi-pass (i.e., GPR77) membrane proteins could be displayed on the envelopes of herpes simplex virus-1 (HSV-1) with correct orientation and conformation[9]. To demonstrate the power of VirD technology in this study, we fabricated a high-content VirD-GPCR array, comprised of 315 non-odorant GPCRs, and demonstrated that the VirD-GPCRs were folded and functional using a variety of biochemical assays.

## Results

### Fabrication of a non-odorant VirD-GPCR array.
To develop the first high-content VirD array, we decided to focus on the 370 non-odorant GPCRs. We assembled a collection of open reading frames (ORFs) encoding 337 non-odorant GPCRs that were available to us. To enable high-throughput cloning of these ORFs into the HSV-1 strain KOS genome carried in a bacterial artificial chromosome (BAC) vector[10], we replaced the glycoprotein B coding sequence (UL27) with a Gateway cloning cassette. The engineered molecule also contained the sequences for the expression of a V5 epitope at the C-terminus of the cloned ORF followed by a STOP codon (Fig. 1a). Meanwhile, STOP codons of the GPCR ORFs were removed via PCR reactions and the modified ORFs subcloned into the Gateway Entry vector. Single colonies were picked, followed by Sanger sequencing to ensure error-free subcloning of each ORF. Confirmed STOP-free ORFs were then shuttled into the UL27 (gB) locus in the HSV-1 genome using LR recombination reactions and transformed into Escherichia coli by electroporation. For each bacterial transformation, at least two colony PCR reactions were performed using a primer pair that annealed to the viral sequences flanking the cloning site of the GPCR ORFs, and gel electrophoresis was employed to examine whether the amplicon was of the expected size of the GPCR cloned. To this end, we have successfully subcloned a total of 332 (98.5%) GPCR ORFs into the UL27 locus.

To produce recombinant viruses, we first transfected each GPCR::ΔUL27 BAC DNA into a Vero transformed cell line, D87, that complements the growth of mutants that do not express gB. When viral plaques became evident, 5–7 days post-transfection, low titer viral stocks were harvested for each GPCR recombinant virus. After a secondary infection, expression of a total of 317 GPCRs was detected in total cell lysates with anti-V5 antibodies (Fig. 1b). Next, high titer stocks of the 317 VirD-GPCRs were individually prepared following infection of D87 cells. Since we observed that expression levels of different GPCRs varied in different cell lines, we prepared the final VirD-GPCR virions from Vero-, HEL-, HeLa-, and HEK-293T-infected cells to maximize the production of VirD-GPCRs (Supplementary Fig. 1a, b). The VirD-GPCR virions were further purified to homogeneity via sucrose cushion and resuspended in a small volume to maintain a high virion concentration. A small fraction of each purified VirD-GPCR virion was subjected to anti-V5 immunoblot (IB) analysis, based on which 315 VirD-GPCRs passed this quality control step (Fig. 1b). Finally, the 315 virion preparations were arrayed into a 384-well titer dish and robotically printed in duplicate onto SuperEpoxy slides to form the VirD-GPCR array. The quality of the printed VirD-GPCR arrays was examined with anti-gD antibody and all of the 315 arrayed VirD-GPCRs showed significant anti-gD signals as compared with the negative controls (e.g., bovine serum albumin (BSA)) (lower left panel; Fig. 1b). Moreover, scatter plot analysis of an anti-gD assay performed in duplicate indicated a high reproducibility with a correlation coefficient of 0.92 (Supplementary Fig. 1c, d). Therefore, we successfully produced a high-content VirD array that covers 85% of the annotated non-odorant human GPCRs.

### Profiling antibody specificity on VirD-GPCR array.
Antibody-based biologicals are emerging as the next-generation therapeutics because of their unique properties, such as superior pharmacokinetics, simple formulation, and modular, easily engineerable format. Indeed, Erenumab (trade name Aimovig) was recently approved by the FDA as the first antibody-based drug that targets the GPCR, i.e., CGRPR, for the prevention of migraine[11]. However, there are serious problems with the quality and consistency of antibodies because of the absence of standardized antibody-validation criteria, a lack of transparency from commercial antibody suppliers and technical difficulties in comprehensive assessments of antibody cross-reactivity[12–17].

To demonstrate that the VirD-GPCR arrays can be used to test antibody specificity, we selected 20 commercially available mAbs

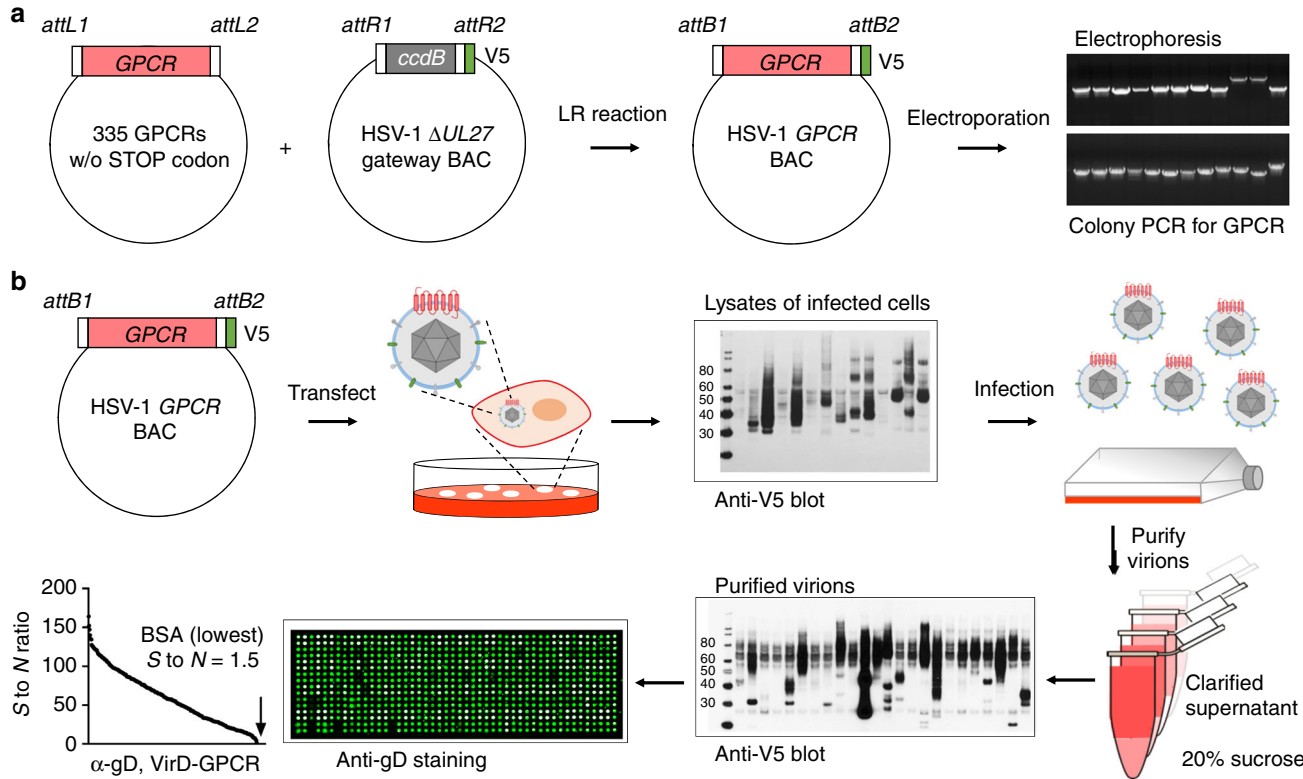

**Fig. 1** Construction of high-content VirD-GPCR array. **a** Subcloning of 335 human GPCR ORFs into the *UL27* locus of the HSV-1 genome. After the STOP codons were removed from the 335 available GPCR ORFs, they were subcloned into the *UL27* locus in the HSV-1 genome on a BAC vector, resulting in fusion with a V5 tag at their C-termini (middle panel). After bacterial transformation, colony PCR reactions were carried out and the products examined using electrophoresis to identify the correct construct (right panel). **b** Production of VirD-GPCR virions and VirD array fabrication. Confirmed recombinant virus constructs were individually transfected to Vero cells and the viruses were harvested ~7 days post-transfection. Anti-V5 mAb was used to examine expression of the GPCRs as a quality control. Passers were next used to infect cells for virion production. After sucrose cushion centrifugation, a fraction of purified virions was examined again with anti-V5. 315 VriD-GPCRs passed this quality control step and were spotted onto a glass slide to form VirD-GPCR array. The quality of VriD-GPCR array was examined using anti-gD mAb, followed by a Cy3-labeled anti-mouse IgG antibody. All the ViP.rD-GPCRs on the array showed positive anti-gD signals while the BSA showed the lowest signals

targeting 20 human GPCRs. The criteria used to select the 20 commercial antibodies were the following: (i) they have to be monoclonal antibodies; (ii) they should recognize ectodomains of the human GPCRs; and (iii) they have to be flow cytometry-positive when used against intact cells (Table 1). We applied each mAb individually to a pre-blocked VirD-GPCR array, measured the corresponding binding signals using fluorescently labeled secondary antibodies, and determined the Z-scores of all the VirD-GPCRs for that mAb on the basis of the standard deviation (SD) value for each assay. Using a stringent cut-off value of Z-score ≥5, only 9 of the 20 tested mAbs recognized their intended targets specifically. Two examples of anti-CXCR2 and -CCR7 are shown in Fig. 2a, b. Of note, four of them are known to have neutralization activity. One other mAb, anti-ACKR3, not only recognized VirD-ACKR3 as its top 1 target, but also showed Z-scores of 6.5 and 5.9 to VirD-CALCR and VirD-GPR61, respectively, suggesting off-target binding activity. The remainder ten mAbs, however, completely failed to show any detectable binding activities to their intended targets, and some of them bound to many unrelated VirD-GPCRs (i.e., anti-DRD1 in right panel; Fig. 2a).

Although an easy explanation is that these mAbs are of poor quality and thus failed in this test, it was also possible that the 11 GPCRs were displayed in the wrong orientation/misfolding in the virions. To explore the latter possibility, we first focused on the off-target binding activity observed with anti-ACKR3 mAb. Using intact Vero cells infected with viruses carrying ACKR3, CALCR,

or GPR61, we performed immunofluorescence assays (IFA) with anti-ACKR3. As shown in Fig. 2c, the mAb showed strong staining to all three infected cells as compared with the negative control cells infected with K082, a *ΔUL27* HSV-1 virus, suggesting that this mAb could recognize CALCR and GPR61 embedded in intact cell membranes. Similarly, this mAb also strongly recognized all three GPCRs in IB analyses against cell lysates of these infected cells under either native or denatured conditions (Fig. 2d). Interestingly, sequence alignment analysis using the ectodomain sequences of the three GPCRs identified a highly conserve 6-mer motif of [NMF]GEL[VTG][RC], suggesting a commonly shared epitope for this mAb. Homology search did not identify any other non-odorant GPCRs that carry a 6-mer peptide with high similarity.

Next, we randomly selected 4 (i.e., anti-DRD1, -CCR2, -CCR9, and -S1PR1) of the 10 mAbs that completely failed to recognize their intended GPCRs on the VirD-GPCR arrays for the IFA and IB analyses. All of the four mAbs failed these tests; anti-DRD1 is shown in Fig. 2e, f as an example (Fig. 2e, f; Supplementary Fig. 2). The failure of these mAbs was not due to poor expression of their target GPCRs because these GPCRs could be readily detected with anti-V5 (Fig. 2f). To determine whether these failed mAbs could recognize linear epitopes, we performed mAb-binding assays on HuProt arrays, each comprised of 20,240 human proteins in full-length and denatured using 9 M urea treatment[18]. Except anti-S1PR1, which was not tested because S1PR1 was not available on HuProt, the rest nine mAbs failed to

**Table 1 Binding specificity for 20 commercial mAbs using VirD-GPCR arrays**

| mAbs | Provider | Assays on the datasheet | GPCR-VirD array results | Z-score |
|---|---|---|---|---|
| α-CCR7 | R&D System | Neutralization; Flow; IFC | Specific | 16.5 |
| α-CXCR1 | R&D System | Neutralization; Flow; IHC | Specific | 9.2 |
| α-CXCR2 | R&D System | Neutralization; Flow; IHC | Specific | 15.6 |
| α-CXCR5 | R&D System | Neutralization; Flow; IFC; IHC | Specific | 11.4 |
| α-GPRC5C | R&D System | Flow | Specific | 16.1 |
| α-ACKR1 | R&D System | Flow; IFC | Specific | 18.1 |
| α-LTB4R | R&D System | Flow | Specific | 13.4 |
| α-SSTR2 | R&D System | Flow; IFC; IHC | Specific | 11.8 |
| α-MRGPRF | R&D System | Flow; WB | Specific | 10.8 |
| α-ACKR3 | R&D System | Flow; IHC | Specific[a] | 14.0 |
| α-DRD1 | R&D System | Flow; IHC | Nonspecific | 2.2 |
| α-CCR1 | R&D System | Flow | Nonspecific | 1.5 |
| α-CCR2 | R&D System | Flow, IHC | Nonspecific | -0.5 |
| α-CCR4 | R&D System | Flow | Nonspecific | 1.4 |
| α-CCR9 | R&D System | Flow; IFC; IHC | Nonspecific | 0.8 |
| α-CXCR6 | R&D System | Flow | Nonspecific | −0.7 |
| α-S1PR1 | R&D System | Flow | Nonspecific | 1.2 |
| α-APLNR | R&D System | Flow; IHC | Nonspecific | 2.4 |
| α-P2RY11 | R&D System | Flow | Nonspecific | 2.7 |
| α-P2RY13 | ProMab | Flow, ELISA, WB | Nonspecific | 1.0 |

Twenty commercial mAbs against 20 GPCR ectodomains were purchased and tested individually on the VirD-GPCR arrays. Z-scores that were calculated for the intended targets of each mAb obtained on the VirD-GPCR arrays are shown. [a]VirD-ACKR3 was recognized as the top target with two off-targets, CALCR and GPR61

recognize their intended targets as the top targets. Similarly, none of them recognized their intended targets under native conditions on HuProt arrays (Supplementary Fig. 3).

**Functional test of GPCR activity with canonical ligands**. To demonstrate further that the ten GPCRs that were not recognized by their respective mAbs were functional/folded correctly, we decided to test their binding activities to their canonical ligands using an imaging approach. Four of the ten ligands were commercially available small molecules with fluorescent labels; the rest were peptide ligands and we labeled them with NHS-conjugated dyes. We immobilized these VirD-GPCRs on a passivated cover slip and incubated with their corresponding ligands at low nanomolar concentrations. Using single-molecule imaging on total internal reflection fluorescence (TIRF) microscopy, we recorded the resulting fluorescent images and compared the binding signals with that of the negative control, K082 virus. Except P2RY13, all the nine VirD-GPCRs that were not recognized by the commercial mAbs showed significantly higher binding signals than the K082 control, indicating that they were functional and folded correctly (Fig. 2g; Supplementary Fig. 4). As an example, $D_1$ antagonist showed significantly higher binding signals to its canonical receptor DRD1 than K082 (Fig. 2g). P2RY13 receptor was not observed to interact with its ligand, ATP, presumably due to the weak affinity in the low micromolar range[19]. Taken together, the high-content non-odorant VirD-GPCR array was validated as a powerful platform to screen for high-quality mAbs against folded GPCRs.

**Specificity test for GPCR ligands**. The success of the above approach prompted us to determine whether VirD-GPCR arrays could be used to examine binding specificity of GPCR ligands. As a proof of concept, we chose two peptide ligands, dynorphin A and somatostatin-14 (SRIF-14), to perform specificity tests. The function of the labeled ligands was largely unchanged based on the calcium signaling measured in Gα-15 cells infected with the recombinant virus expressing the GPCR (Supplementary Fig. 5).

As illustrated in Fig. 3a, dynorphin A bound to its canonical receptor, OPRD1, as the top receptor with significant signal

intensity[20]. To demonstrate that dynorphin A can activate OPRD1, we performed calcium imaging assays in OPRD1-infected cells and showed that dynorphin A could indeed induce $Ca^{2+}$ influx, although to a lesser extent than OPRK1-infected cells (Supplementary Fig. 5). To better understand why OPRK1 did not show detectable binding signals to dynorphin A on the VirD-GPCR arrays, we measured its expression level in purified viruses using anti-V5 antibodies and found that it was present at a very low level (Supplementary Fig. 6a). To rule out the possibility that the kappa receptor was not correctly folded in the envelope of the virions, we employed a more sensitive TIRF microscopy method and demonstrated that VirD-OPRK1 showed significantly higher binding signals to Cy3-labled dynorphin A than the K082 control virions (Supplementary Fig. 6b). Taken together, this new evidence confirmed that dynorphin A could activate both kappa and delta receptors in cells and that virion-displayed kappa receptor could bind to dynorphin A.

SRIF-14 bound to more than 15 VirD-GPCRs with Z-scores ≥2 (Fig. 3b), including its known receptor, SSTR2 (ref. [21]). To determine whether the observed off-target binding activities of SRIF-14 were not due to an artifact caused by dye-labeling, we employed a cell-based competition assay to examined binding specificity between SRIF-14 and three randomly selected off-target GPCRs, namely GABBR2, NTSR1, and KISS1R, with Z-scores ≥2 (Fig. 3c). The three VirD-GPCR constructs were used to separately infect Vero cells and VirD-SSTR2 and K082 were also included as positive and negative controls, respectively. The same fluorescently labeled SRIF-14 was added to these infected Vero cells in the absence or presence of cold SRIF-14 (i.e., agonist) or cyclosomatostatin (i.e., antagonist). Quantitative analysis clearly showed that all cell lines infected with the three off-target GPCRs showed significantly higher binding signals than K082-infected cells (upper panel of Fig. 3c, d), comparable to those infected with SSTR2. More importantly, both cold SRIF-14 and cyclosomatostatin (cycloSST) could readily compete off the binding signals of Cy5-labeled SRIF-14 on the cells infected with the three off-target GPCRs, suggesting ligand-specific interactions (middle and lower panels of Fig. 3c, d).

Because the array- or cell-based assays can only offer an end-point measurement, we decided to obtain true binding kinetics

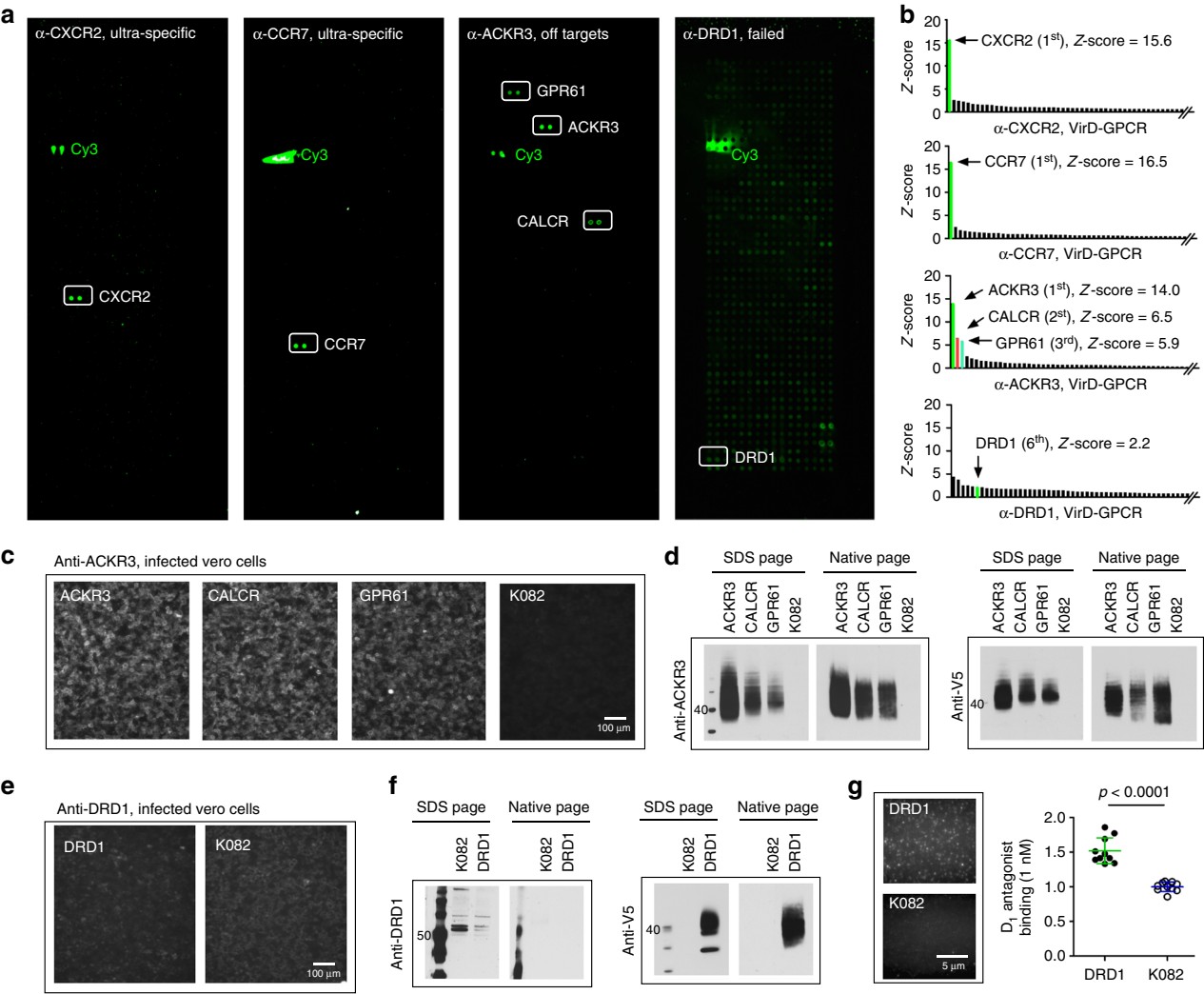

**Fig. 2** Specificity tests of commercial mAbs on VirD-GPCR arrays. **a** Examples of binding signals obtained with commercial mAbs. Anti-CXCR2 and -CCR7 are shown as ultra-specific; anti-ACKR3 can cross-react with CALCR and GPR61; anti-DRD1 completely failed to recognize its target while showing nonspecific binding activities to other GPCRs. **b** Histograms of Z-scores obtained with three mAbs. Z-scores of the two off- targets identified by anti-ACKR3 are also shown. **c** Immunofluorescence analysis (IFA) validation of anti-ACKR3 to its off-targets in infected Vero cells. K082-infected cells are shown as a negative control. **d** Immunoblot analysis also confirmed that anti-ACKR3 can recognize its two off-targets in the cell lysates of infected Vero cells under both denatures and native conditions. **e**, **f** Anti-DRD1 failed to recognize DRD1 in VirD-DRD1-infected cells using IFA (**e**) or immunoblot (IB) analyses (**f**). **g** Single-molecule imaging using TIRF microscopy to determine interactions between VirD-DRD1 and its canonical ligand D$_1$ antagonist. K082 virions were used as a negative control. Quantitative analysis of TIRF imaging demonstrated that VirD-DRD1 showed significantly higher binding signals to D$_1$ antagonist than K082. Data were analyzed with two-tailed Student's t-test, $n = 10$

using a previously reported virion-oscillator approach[22–24]. It is a label-free plasmonic imaging technique that can quantify the ligand-binding-induced mobility change of the virions anchored on the surface of the sensor chip via a flexible molecular linker and reveal the binding kinetics of small-molecule ligands. VirD-GABBR2, -NTSR1, and -KISS1R were separately attached to a gold-coated cover slip via a flexible polyethylene glycol (PEG) linker (Fig. 4a). Again, VirD-SSTR2 and K082 were included as the positive and negative controls, respectively. Using surface plasmon resonance (SPR) to monitor voltage-induced oscillation of the VirD-GPCRs, we recorded oscillation amplitude changes of VirD-GPCRs as a result of association and dissociation of SRIF-14 in a real-time, label-free fashion. As illustrated in Fig. 4b, f, all four tested VirD-GPCRs showed typical association and dissociation curves in a dose-dependent manner, which allowed us to determine the corresponding $k_a$ and $k_d$ values (Supplementary Table 1). As expected, K082 did not show any detectable binding

kinetics even in the presence of 8 μM SIRF-14 (Fig. 4f). Binding affinity $K_D$ values were thus calculated for SSTR2, GABBR2, NTSR1, and KISS1R to be 11.2 nM, 0.4 μM, 2.6 μM, and 25.0 μM, respectively. These independent experiments confirmed the off-target-binding activities of SIRF-14 observed on VirD-GPCR arrays.

**Discovery and characterization of GPCR targets for GBS**. In recent years, several elegant studies reported that both Gram-negative and -positive bacterial pathogens, such as *Neisseria meningitidis* and *Streptococcus pneumoniae*, could utilize human GPCRs (e.g., ADRB2 and PTAFR) as receptors to penetrate human epithelial cells[25,26]. *Streptococcus agalactiae* (a.k.a. group B *Streptococcus* or GBS) is the most common Gram-positive organism causing neonatal meningitis by penetrating human blood–brain barriers (BBBs). However, its host receptor has remained elusive. To explore the possibility that GBS may exploit

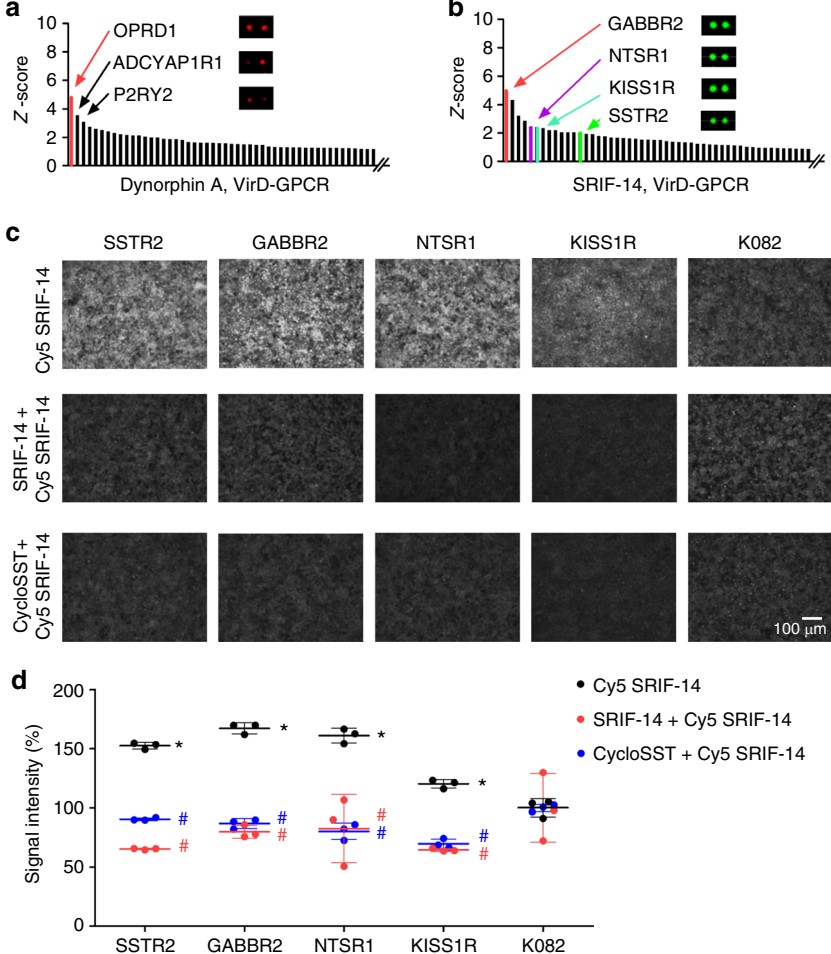

**Fig. 3** Identification and cell-based validation of peptide ligand–GPCR interactions. **a** Commercially available Dynorphin A was Cy5-labeled and probed to a VirD-GPCR array. Quantitative analysis showed that it bound to VirD-OPRD1 with the highest Z-score followed by ADCYAP1R1 and P2RY2. **b** A commercially available peptide SRIF-14 was Cy3-labeled and probed to a VirD-GPCR array. Quantitative analysis revealed that it bound to several unexpected off-targets in addition to its canonical receptor, SSTR2. **c** Vero cells were separately infected with SSTR2, GABBR2, NTSR1, KISS1R, or K082 virus. Infected cells were then incubated with Cy5-labeled SRIF-14 at 8 μM in the absence (upper panel) or presence of cold SRIF-14 (middle panel) or cyclosomatostatin (cycloSST, lower panel). **d** Quantitative analysis of binding signals. Each binding assay was performed in triplicate and the obtained binding signals were normalized to those of the K082 controls. Data were analyzed by two-way ANOVA with repeated measures followed by Bonferroni post-test. *$P < 0.05$, comparison between VirD-GPCRs and K082 in the absence of competitor ligands; #$P < 0.001$, comparison between binding signals obtained in the absence and presence of the competitor ligands. $n = 3$, biologically independent samples

host GPCRs as a means to penetrate BBB, we probed the VirD-GPCR array with fluorescently labeled live GBS K79 (a strain isolated from a neonate with meningitis) in duplicate to discover potential host receptors. As a comparison, a Gram-positive non-pathogenic bacterium, *Streptococcus gordonii*, which does not penetrate BBB, was used as a negative control. Five VirD-GPCRs, namely GPR101, GPR148, LHCGR, CysLTR1, and LGR5, showed significantly higher binding signals to K79 than *S. gordonii* in a reproducible manner (Fig. 5a).

Of the five identified potential GBS receptors, GPR101, GPR148, and LGR5 are orphan receptors without identified canonical ligands. LHCGR is mainly expressed in ovary and testis and binds luteinizing hormone; mutations in this gene are known to cause infertility[27]. Therefore, these four candidates are either difficult to pursue or less relevant to the pathogenesis of meningitis. The last candidate receptor, CysLTR1, recognizes cysteinyl leukotrienes and is an attractive candidate for several reasons. First, its activation is associated with increased permeability of BBB, as well as promotion of the movement of leukocytes from the blood into brain tissues in animal models[28].

Second, its activation may also increase the entry of leukocyte-borne viruses, such as HIV-1, into brain tissue[29]. Third, a well-established antagonist, Montelukast, specifically inhibits CysLTR1 but not its homolog CysLTR2 (ref. [30]). Importantly, recent studies have demonstrated that Montelukast could protect against hippocampus injury induced by transient ischemia and reperfusion in rats[31].

To validate further the discovery of CysLTR1 as a potential target of GBS, we first employed cultured human brain microvascular endothelial cells (HBMEC) to evaluate the role of endogenous CysLTR1 in GBS penetration. HBMEC are the major component of the BBB and CysLTR1 is known to be expressed in this cell line[32]. Monolayers of HBMEC were pretreated with Montelukast at different concentrations for 1 h, washed, and infected with GBS K79 (see Methods for more details). DMSO-treated cells were used as a vehicle control. After removal of unbound bacteria with several washes, antibiotics (gentamicin and penicillin) were added to the cells to kill extracellular bacteria. To evaluate GBS penetration to the cells under different conditions, cells were lysed, diluted, and plated onto blood agar

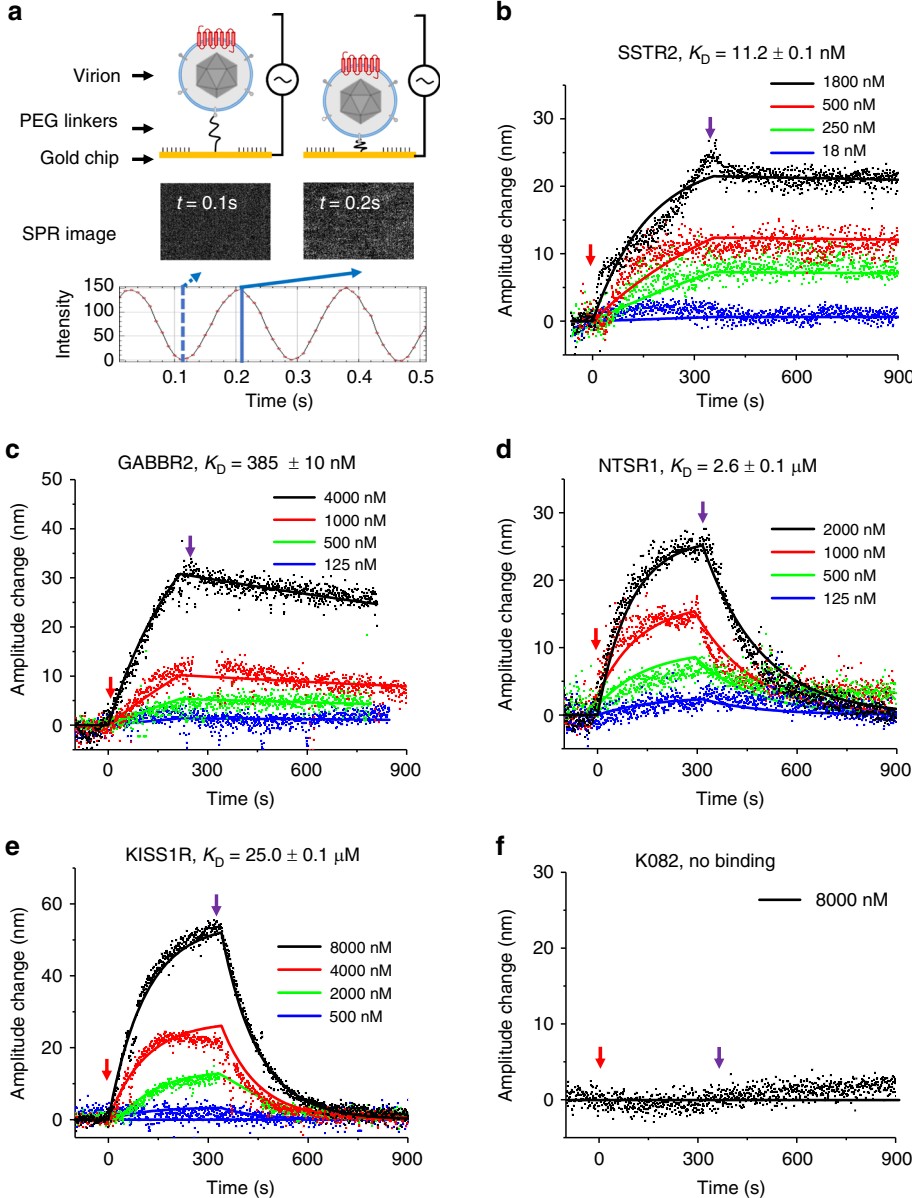

**Fig. 4** Binding kinetics of SRIF-14 to SSTR2, K082, and other three virions. **a** Principle of binding kinetics measurement using virion-oscillator device. **b–f** Binding kinetics of SRIF-14 to its canonical receptor, SSTR2, and three newly discovered off-target GPCRs. The red and purple arrows mark the starting time points of the association and the dissociation phases, respectively. Binding curves were fit using the first order kinetics model (solid lines). The calculated affinity values range from 11.2 nM (SSTR2) to 25.0 μM (KISS1R), while K082 virion showed no detectable binding activity to the SRIF-14. Detailed $k_a$, $k_d$, and $K_D$ values are listed in Supplementary Table 1

plates. Colonies formed on the agar plates were counted and used as a proxy to evaluate GBS penetration of the BBB (see Methods). After normalization to the DMSO controls, it was clear that Montelukast significantly inhibited GBS penetration to HBMEC in a dose-dependent fashion. In fact, as much as 77% GBS penetration was inhibited by 50 μM Montelukast (Fig. 5b). These results corroborated our VirD-GPCR array results that CysLTR1 might play an important role in GBS penetration of the BBB.

To further demonstrate the role of CysLTR1 in GBS penetration into the brain in vivo, we employed a mouse model of experimental hematogenous meningitis (Fig. 5c). A group of six mice were each intraperitoneally administered Montelukast (5 mg kg⁻¹), while another group of five mice received DMSO as a vehicle control. After 2 h each mouse received $1 \times 10^8$ CFU of GBS (K79) via the tail vein injection. One hour later, blood was

collected and plated for bacteria counts. Immediately following the blood collection, mice were transcardially perfused to remove the remaining body blood and the brains were removed, weighed, homogenized, and plated for bacterial counts (see Methods for more details). We found that the administration of Montelukast reduced the GBS infection in the mouse brains by an average of 81% as compared to that of the DMSO controls. It is important to note that no significant differences of GBS counts were observed in the blood between the two groups, indicating that decreased GBS penetration into the brain was not the result of having less bacterial counts in the blood at the time of collecting the brain specimens (Fig. 5c). Taken together, these experiments demonstrated that using the VirD-GPCR array as an unbiased screening platform, we successfully identified CysLTR1 as a receptor for GBS penetration into host cells both in vitro and in vivo.

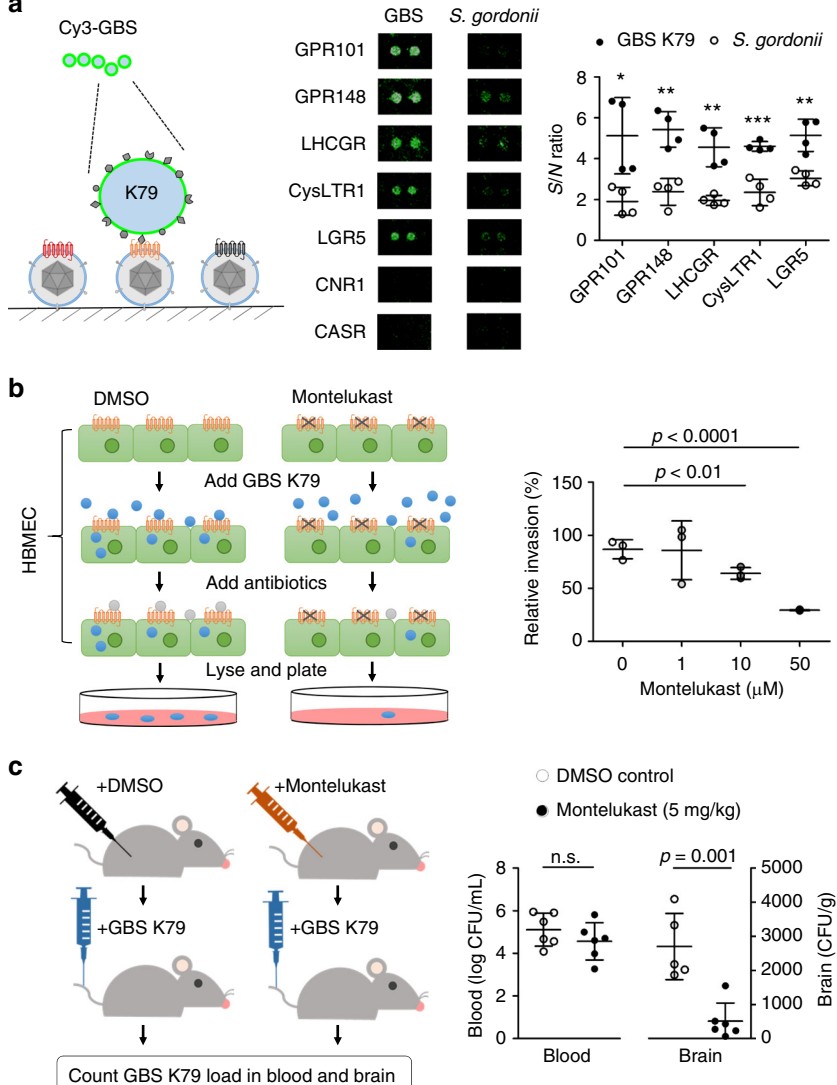

**Fig. 5** Discovery and validation of a new GPCR receptor for GBS. **a** Five VirD-GPCRs were identified as potential candidate receptors. A clinical strain K79 of GBS was Cy3-labeled and probed to a VirD-GPCR array. In parallel, *S. gondonii* was used as a non-pathogenic negative control (middle). CNR1 and CASR were not bound by either GBS or *S. gondonii*. Quantitative analysis of the binding signals from the two bacteria identified five GBS-specific GPCRs (right). $n = 2$, biologically independent samples. **b** In vitro validation of CysLTR1. Human brain microvascular endothelial cells (HBMEC) were pretreated with Montelukast to block CysLTR1, followed by incubation with GBS. After washes, antibiotics were added to kill free bacteria and HBMEC were lysed and plated onto blood agar plates. After overnight incubation, numbers of GBS colonies were counted. As compared with the DMSO-treated negative control, Montelukast showed a dose-dependent inhibition of GBS invasion into HBMEC. $n = 5$, biologically independent samples. **c** In vivo validation of CysLTR1. A group of mice was each intraperitoneally administered Montelukast ($n = 6$) or DMSO ($n = 5$). After 2 h each mouse received $1 \times 10^8$ CFU of GBS (K79) via the tail vein injection. One hour later, blood and homogenized brains were collected and plated for bacterial counts. Using the same colony formation method, administration of Montelukast reduced GBS brain infection in the mice by an average of 81% as compared to the DMSO controls (right). No significant differences in the levels of GBS counts were observed in the blood between the two groups. Data were analyzed with two-tailed Student's *t*-test whereas *$P < 0.05$, **$P < 0.01$, and ***$P < 0.001$

## Discussion

The VirD method was developed to display membrane protein on the lipid envelope of HSV-1 virion. To take a full advantage of VirD technology, we built a high-content non-odorant GPCR array, comprising 315 human GPCRs. This is the first array of its kind, covering >85% of all annotated non-odorant GPCRs, which are preferred drug targets. Using this high-content VirD-GPCR array as a tool, we were able to characterize the specificity of commercial antibodies and canonical ligands, and to identify new receptors involved in host–pathogen interactions.

Unlike the existing display systems, such as liposomes, nanodiscs[33,34], and cell membrane fractions[7,8], GPCRs displayed

in the VirD-GPCR system are most likely to orientate uniformly with preserved conformation, as demonstrated in this study and Hu et al.[9]. In addition, copies of the GPCR molecules displayed on a single virion are more uniform because they are expressed as viral proteins and assembled by the virus machinery. Another display technology generates virus-like particles (VLPs) to maintain native conformation of membrane proteins[35]. VirD-GPCRs carry genetic information and infect almost every cell at high MOI. In addition, VirD-GPCR can maintain its infection efficiency in many mammalian cell lines[36] and generate non-infectious virion due to lack of gB[37]. Unlike VirD, however, VLPs do not carry any genetic information and are not self-renewable.

Therefore, transient transfection is needed every time to regenerate VLPs, raising concerns of transfection efficiency (40–90%) and reproducibility[35,38,39]. Therefore, VirD-GPCRs are biologically safe and more amenable to large-scale production.

Another unique advantage of VirD technology that has not yet been fully exploited is the flexibility in choice of cell lines for VirD production, since HSV-1 can infect many cell line in vitro[36]. For example, we have used HEL, Vero, HeLa, and HEK-293T cells for virion production to obtain the optimal GPCR expression (Supplementary Fig. 1a, b). This can be very important for GPCRs that require tissue-specific post-translational modifications (e.g., glycosylation) to maintain their native functions. Moreover, the infectious stocks of the VirD-GPCR collection generated in this study can be readily used to infect many primate cell lines, facilitating downstream mechanistic studies, such as calcium imaging and β-arrestin recruitment. In its current form, a single GPCR is displayed per virion. However, many GPCRs are known to form obligate heterodimers (e.g., GABA receptor GABBR1/GABBR2) or even multimers to achieve additional regulation[40,41]. Because of the flexibility of the VirD platform, we can co-infect human cells with more than one type of VirD-GPCR to produce chimeric heterodimer or multimer VirD-GPCRs.

Monoclonal antibodies (mAbs) are fast growing therapeutic modalities and account for ~30% of the new drugs approved by the FDA in recent years[42,43]. Indeed, FDA has just approved the first mAb-based drug targeting CGRPR[11]. However, mAb specificity has been one of the major concerns and antibodies of poor performance have wasted about half of the spending on the protein-binding reagents[15–17]. This is particularly true for the GPCR community because a major hurdle to generating antibodies against GPCRs is obtaining folded GPCRs as antigens for animal immunization or display platform-based discovery. Since the conformation of VirD-GPCRs is preserved, VirD-GPCR arrays offer a high-throughput platform to screen for functional antibodies of high specificity with therapeutic potentials. Indeed, we purposefully selected 20 commercial mAbs that were claimed by the manufacturers to recognize the ectodomains of 20 different human GPCRs. However, only nine mAbs turned out to be ultra-specific, four of which are known to have neutralization activities, indicating that they recognize functional epitopes on VirD-GPCR arrays. Although 10 commercial mAbs failed to recognize their intended targets in VirD-GPCR-binding assays, 9 of the 10 VirD-GPCRs could bind their canonical ligands as demonstrated with TIRF microscopy, indicating that these GPCRs are all folded correctly. Therefore, we believe that VirD-GPCR array is an ideal platform to identify the specificity and potential therapeutic value of mAbs against native and fully folded GPCRs.

Another application of the VirD-GPCR array is to profile the specificity of ligands. Cy5-labeled Dynorphin A bound to one of its canonical receptors on the VirD-GPCR array with high specificity, but Cy3-labeled SRIF-14 bound to multiple unanticipated receptors (e.g., GABBR2, NTSR1, and KISS1R) in addition to its canonical receptor, SSTR2. These off-target binding activity was further validated using the virion-oscillator technology, even though the binding affinity of SRIF-14 to the off-target GPCRs was in the low μM to high nM range. Peptide ligands of low affinity are a common property of many GPCRs, such as GPR55 binds to cannabinoid[44]. However, detection of binding activity is only the first step to identify a new ligand for GPCRs; in vitro and in vivo functional assays, such as calcium imaging, PRESTO-Tango assays, and animal-based studies, will be required to fully characterize these new interactions. Nevertheless, VirD-GPCR arrays can offer a rapid, comprehensive survey for specificity of ligand–GPCR interactions.

Although GPCRs are mostly known for their involvement in a wide variety of physiological processes, such as inflammation, neurotransmission, and autonomic nervous system transmission, recent studies also demonstrated that some GPCRs play an important role in pathogen–host interactions[25,26]. GPCRs can serve as an entry point for pathogens because GPCRs undergo an internalization process via β-arrestins following activation. To explore the possibility of using the VirD-GPCR array to identify novel receptors for bacterial pathogens, we probed labeled GBS on a VirD-GPCR array and readily identified several GPCR candidates that showed specific interactions as compared with a negative control bacterium. By focusing on one candidate receptor, CysLTR1, we demonstrated that this newly identified GPCR indeed played a crucial role in GBS invasion of BBB both in vitro and in vivo. We believe that this approach can be easily applied to many other microbial pathogens to better understand the molecular mechanisms of pathogen invasion and provide novel therapeutic targets.

## Methods

**Cells**. Vero cells (ATCC CCL-81), the gB complementing Vero cell line (D87), Human Embryonic Lung (HEL, ATCC CCL-137), HeLa (ATCC CCL-2), and HEK-293T (ATCC CRL-3216) cells were grown in minimum essential medium supplemented with 10% fetal calf serum (Gibco-Invitrogen). D87, the gB complementing cell line was generated previously[45] and used to propagate all HSV-1 GPCR expressing viruses. Cell lines were passaged 2–3 days in Dulbecco's modified Eagle's medium supplemented with 10% fetal bovine serum[46]. HBMEC were isolated and characterized[47]. HBMEC were grown in RPMI 1640 containing 10% fetal calf serum, and 10% Nu-Serum in a 37 °C incubator with 5% $CO_2$ until reached confluence.

**Antibodies**. mAbs reactive to ectodomain of human GPCRs were purchased from R&D system (MAB197, MAB330, MAB331, MAB190, MAB6594, MAB4139, MAB099, MAB4224, MAB8396, MAB42273, MAB150, MAB179, MAB8276, MAB2016, MAB8561, MAB145, MAB1567, MAB699, and MAB9305) except for P2RY13 which was from ProMab (30487). Anti-HSV-1 gD antibody DL6 was purchased from Santa Cruz Biotechnology (sc-21719) and ascities fluid for DL6 was a kind gift from David Johnson (OHSU). Anti-V5 antibody was purchased from Thermo Fisher Scientific (R960-25). Cy3-conjugated anti-mouse IgG was purchased from Jackson ImmunoResearch.

**Engineering HSV-1 strain KOS for Gateway cloning**. The HSV-1 strain KOS genome cloned into a BAC plasmid was kindly provided by David Leib (Dartmouth University)[10]. This genome (KOS-37 BAC) was purified from the E. coli host using the Purelink DNA purification kit (Invitrogen) and 1 μg electroporated into E. coli ccdB Survival™ 2 T1R strain (Thermo Fisher Scientific). Recombination of the Gateway reading frame A cassette into this genome was carried out using the Gene Bridges Red/ET recombination kit (Gene Bridges) and the protocol provided. We first electroporated plasmid pSC101(20 ng) into the ccdB survival strain harboring KOS-37 BAC and selected for colonies on chloramphenicol and tetracycline plates. Positive clones containing both the KOS-37 BAC and pSC101 plasmids were made electrocompetent and transformed with a PCR fragment that encodes the Kanamycin gene (1.4 μg). This gene was PCR amplified from plasmid rpsL-kanR using primers that contain homology to the UL27 gene locus (F-TCCAGCACC TCGCCCCCAGGCTACCTGACGGGGGGGCACGACGGGCCCCCGTAGTCC CGCCGGCCTGGTGATGATGGCGGGGATCG, R-AAACCAAAAGATGCGCA TGCGGTTTAACACCCGTGGTTTTTTATTTACAACAAACCCCCCGTCAGAAG AACTCGTCAAGAAGGCG). Colonies were selected on chloramphenicol/tetracycline/kanamycin agar plates. Positive clones were screened by sequencing for the correct replacement of the UL27 ORF with that of the kanamycin gene as well as DNA that could give rise to virus following transfection in D87 cells. This virus genome was designated KΔUL27Kan. Subsequently, the counter selection was performed by introducing the UL27-GWA rfa CV5 cassette into the ccdB strain containing KΔUL27Kan. This PCR product was amplified using primers with homology to UL27 sequences and to the Gateway sequence in plasmid pDEST42 which encodes gateway reading frame A and a C-terminal V5 sequence (Invitrogen) (F- ACGACGGGCCCCCGTAGTCCCGCCCATCACAAGTTTGTACAAA AAAGCTG, R-TTTTTATTTACAACAAACCCCCCGCCGCTACGAGAATCGA GACCGAGGAGAG). Electroporation of the PCR product and subsequent recombination was performed as described in the manufacturer's protocol. The colonies were plated on chloramphenicol and streptomycin selective media and single colonies were screened using PCR assays for the presence of the Gateway sequence. Those that were positive were grown up and the DNA purified from these cells was sequenced to confirm the correct amplification and insertion of the Gateway-CV5 sequences. Further confirmation was also done by transfecting the viral genomes into D87 cells to reconstitute infectivity. This virus genome, designated KOS-GWACV5 served as the destination plasmid for the cloning of all 335 GPCRs.

**Gateway cloning of non-odorant GPCRs**. We assembled a collection of ORFs of 337 non-odorant GPCRs that were available to us. The majority of the GPCR ORFs had a STOP codon, which would prevent fusing the V5 tag to the C-termini of the displayed GPCRs. Hence, we eliminated all the STOP codons via PCR reactions. *E. coli* harboring these ORF plasmids were cultured in 96-well plates at 37 °C for 16 h with 240 r.p.m. rotation. Plasmids were extracted with alkaline lysis using the standard protocol[48]. PCR assays using Phusion high-fidelity polymerase was used to add *attB1/attB2* sites and remove the STOP codons. PCR products with the correct size were gel extracted and purified with QIAquick Gel Extraction Kit (Qiagen). For the BP reaction, 120 ng of PCR product was mixed with 120 ng of pDONR221, 0.6 μL of BP Clonase, and 1 μL of TE buffer for 1 h at room temperature. The reaction mixture was transformed into DH5α and plated onto kanamycin agar plates. Single colonies were grown in 2× YT with kanamycin, and the purified entry plasmid was subjected to *Bsr*GI digestion and sequence confirmation (M13-F and M13R-pUC sequencing primers).

**Cloning GPCR ORFs into the HSV-1 BAC**. The entry plasmids with the correct sequence were used for the LR reactions. For the LR reaction, 50 ng of entry plasmid was mixed with 200 ng of KOS-GWACV5 DNA, 1 μL of LR Clonase, and 2 μL of TE buffer and incubated overnight at room temperature. To make electrocompetent cells, TOP10 cells were overnight cultured at 37 °C, expanded into 100 mL of 2× YT until OD$_{600}$ 0.4–0.6, and washed several times with cold sterile water. The LR reaction was added to the electrocompetent cells, transferred to the electroporation cuvette, and an electrical pulse (1500 V, 25 μF, 200 Ω, and 1 mm gap) was used. The cells were allowed to recover in 2× YT for 90 min, and plated on the chloramphenicol plates. Next day, two single colonies were streaked out to inspect the GPCR inserts using Phire colony PCR and liquid cultures of the positive clones were grown up for BAC DNA isolation.

**Reconstitution of GPCR HSV-1 recombinant virus**. Each GPCR HSV-1 BAC was purified using alkaline lysis methods and alcohol precipitation. This DNA (3 μL) was added to 180 μL OPTI MEM (Thermo Fisher Scientific), mixed with 3 μl X-tremeGENE HP transfection reagent (Sigma-Aldrich), and incubated for 30 min at room temperature. The mixture was then added dropwise into 12-well tissue culture plates that were seeded with D87 cells for 75% confluency. The cells were incubated for 3–9 days to allow virus amplification and then harvested and this low titer stock was designated the Transfection Stock. To confirm that each GPCR was expressed during infection, we performed immunoblots on infected cell lysates using the V5 antibody. Positive viruses were further amplified in D87 cells to produce high titer Working Stocks, which were usually $10^8$–$10^9$ PFU mL$^{-1}$. The working stocks were stored in 10% glycerol at −80 °C.

**Production of HSV-1 virions displaying GPCRs**. In most cases Vero cells were infected with the recombinant HSV-1 virus expressing the human GPCR to produce extracellular virions, which incorporate the GPCR and thus display it on the virion envelop. For some experiments we also used HEL, HeLa, or HEK-293T cells to produce GPCR displaying HSV-1 virions. We observed for some GPCRs, expression of the GPCR in Vero cells was low or absent. This expression could be recovered if HeLa or HEK-293T cells were used for the infection. Cells were cultured within 100 mm tissue culture dishes, and when 100% confluent were infected at an MOI of ~5 PFU per cell in phosphate-buffered saline (PBS) for 1 h. The virus inoculum was removed after the absorption period and growth medium (4 mL per dish) was added and the cells incubated for 48 h. The supernatant was harvested at this time, clarified by centrifugation at 3000 × g for 30 min, overlaid onto 1 mL of 20% sucrose in MEM, and centrifuged at 24,000 × g for 1 h in 4 °C. The pellet from the first sucrose cushion was resuspended with 200 μL cold PBS and sonicated for 15 s. Of this, 5 μL was taken to examine the V5 expression in the virions. The rest was subjected to a second sucrose cushion, 195 μL was layered on top of 200 μL 20% sucrose, centrifuged at 18,000 × g for 1 h in 4 °C, and resuspended in 20 μL PBS + 30% glycerol. These concentrated virions were stored at −80 °C prior to printing.

**Fabricate VirD-GPCR microarray**. Concentrated virions were rearrayed into a 384-well format, printed onto Epoxy slides with duplicate manner with non-contact printer—Arrayjet, immobilized overnight, and kept in −80 °C until usage. The whole printing and immobilization were maintained in 4 °C. On the VirD-GPCR array each spot consisted 800 pL and contained ~$10^4$ virions. K082 (gB null) virion was printed as a negative control.

**Antibody assays on VirD-GPCR arrays**. VirD-GPCR array was thawed in PBS, washed for 5 min, and blocked by 3% BSA for 1 h with 50 r.p.m. shaking. After 5 min wash, arrays were incubated with primary antibodies anti-gD (5000× dilution) or other mAbs against GPCR ectodomains (~ 500× dilution) for 1 h with 50 r.p.m. shaking and then washed with PBS for 5 min × 3. Then, arrays were incubated with Cy3-labeled anti-mouse (1000× dilution) for 1 h with 50 r.p.m. shaking and then washed with PBS + 0.01% Tween-20 for 5 min × 3. There arrays were rinsed with water, centrifuged to dry, scanned by GenePix 4000B, and acquired the intensity of foreground minus background by GenePix software.

**Western blot analysis**. To analyze the expression of each GPCR using the V5 C-terminal fusion tag, we infected Vero cells ($4 \times 10^6$ in a 12-well tray). At 24–48 h post-infection the cells were washed with PBS, lysed with RIPA buffer, clarified the cell lysate, and mixed with an equal volume of 2× Laemmli buffer. To analyze the GPCR-V5 expression in virions, we mixed 5 μL virion suspension from the first sucrose cushion with an equal volume of 2× Laemmli buffer. The mixtures were loaded into NuPAGE (4–12% Bis-Tris; Invitrogen) gels and the proteins resolved in MES running buffer. The proteins in the gels were transferred to Nitrocellulose membranes using the iBLOT 2 gel transfer machine and the membrane blocked in 5% BSA for 1 h at room temperature. The membranes were incubated with anti-V5 antibody (1000× dilution) or anti-gD antibody (5000× dilution) in TN plus Tween (recipe) for 1 h at RT, washed 10 min × 3 in the same buffer, incubated with HRP conjugated anti-mouse (5000× dilution) for 30 min at RT, washed 10 min for four times, incubated with ECL for 2 min, and detected by an X-ray film.

**Immunofluorescence analysis**. Vero cells ($8 \times 10^6$) were cultured in 96-well tissue culture trays and infected with HSV-GPCR virus at an MOI ~10 for 16 h. Infected cells were blocked in 3% BSA PBS for 1 h, incubated with mAb against GPCR ectodomain (500× dilution) for 1 h, washed with PBS 3×, stained with Cy3-conjugated anti-mouse (1000× dilution) for 1 h, washed three times in PBS, fixed with 4% paraformaldehyde in PBS for 20 min, and images were acquired using the Glomax (Promega) fluorescent reader or fluorescence microscopy (Keyence).

**Ligand-binding assays on VirD-GPCR arrays and infected cells**. Dynorphin A and SRIF-14 peptides were purchased from AnaSpec. Dynorphin A (800 μM) and SRIF-14 (300 μM) were labeled with five-fold excess of Cy5- or Cy3-NHS for 1 h at room temperature and quenched by 1 M Tris. The array was blocked with 3% BSA PBS and incubated with Cy5-Dynorphin A (8 μM) or Cy3-SRIF-14 (8 μM) in HEPES buffer for 1 h on a shaking platform set at 50 r.p.m. The array was then subjected to three 5-min washes in cold HEPES buffer, rinsed with water, centrifuged at low speed to remove any moisture, and scanned.

Vero cells ($8 \times 10^6$) were cultured in 96-well tissue culture trays and infected with HSV-GPCR virus at an MOI ~10 for 16 h. Infected cells were blocked in 3% BSA PBS for 1 h, incubated with SRIF-14 (8 μM) in the absence or presence of cold SRIF-14 (800 μM) or cyclosomatostatin (800 μM; Sigma-Aldrich). Cells were washed with PBS 3×, fixed with 4% paraformaldehyde in PBS for 20 min, and images were acquired using the Glomax (Promega) fluorescent reader or fluorescence microscopy (Keyence).

**Single-molecule imaging for ligand-binding measurement**. Borosilicate cover glass was cleaned and activated by air plasma at 300 mTorr for 5 min, followed by dipped in methanol containing 1% *N*-(2-aminoethy)-3-aminopropyltrimethox-ysilane and 5% glacial acetic acid to create amine group on the surface. Cover glass was incubated with epoxide-conjugated polyethylene glycol in Milli-Q water (4.3 mg L$^{-1}$ epoxide-PEG-epoxide) overnight at room temperature and washed with Milli-Q water. The epoxide molecule allows us to link the amine group of virion envelope proteins to the surface, whereas the PEG molecule prevents surface from nonspecific ligand binding and free dye. The microfluidic device was attached to the cover glass with oxygen plasma (100 mTorr) to carry out virion incubation, washing, and buffer exchange. Two channels were built on a single microfluidic device and the volume of each channel was less than 1 μL. One channel of the microfluidic device was incubated with one GPCR target (3 μL) and negative control, K082 (3 μL) overnight at 4 °C. The virion conjugation reaction was then quenched and blocked with PBS-BSA buffer (PBS at pH 7.4 with 0.05% Tween-20, and 0.1 mg mL$^{-1}$ BSA). Peptide ligands were purchased from AnaSpec and labeled with fluorescence as describe above. Cy5-labeled ATP was purchased from Jena Bioscience (NU-805-CY5). The fluorescent D1 antagonist (SKF83566) was purchased from Hello Bio (HB1863). While dye-labeled ligands were incubated in both channels, the images were taken on a TIRF microscope at 50–100 ms exposure time. The 100 nm-height evanescent wave was produced by total reflection of incident light right above the cover glass on a TIRF microscope, then exciting dye-labeled ligands within wave range. The evanescent wave dramatically reduced background derived from excitation of dye molecules in the solution. On bound status, individual dye molecules would be excited and then either photobleached or dissociated from the VirD-GPCRs. Binding dynamics of single ligand and VirD-GPCR was monitored over time in one channel and count individual fluorescent spots from average of time stream images as binding events. For some ligands with either excess free dyes or higher nonspecific binding property, integrated intensity was measured as binding signals instead. To normalize the amount of virions on both channels, the amount of virion conjugated on the surface was quantified with the anti-glycoprotein antibodies. The final binding signals were analyzed by Student's *t*-test and K082 control was normalized to 1.

**Fabrication of virion-oscillators**. The virions were mixed with HS-PEG10k-NHS at 1:5 ratios in 1× PBS and incubated overnight to allow assembly of PEG linkers to the virions via NHS-amine reaction. Then 1 μL of the virion-PEG complex solution containing ~$10^5$ virions was dotted onto a gold chip (47 nm gold on glass) with a pipette. The gold chip was immediately placed in a sealed box with water vapor to prevent drying out the sample. After overnight incubation, the gold chip was rinsed

with 1× PBS to wash off any unbound virion-PEG complex on the surface, and then incubated in 15 nM dithiolalkanearomatic PEG6-COOH spacer for 12 h to passivate the gold surface. Note that all of the modifications above were performed at 4 °C.

**Kinetic measurement of ligand binding**. The protocol was modified based on our recent publication[24]. The plasmonic imaging setup used was a SPRM 200 (Biosensing Instrument), and the images were recorded at 98 frames per second. A sinusoidal potential ($f = 5$ Hz) was applied to the gold chip to drive the virion-oscillators into oscillation. The modulation was applied via a three-electrode system, where the gold chip is the working electrode, an Ag wire is the quasi reference electrode, and a Pt coil is the counter electrode. The virion spot on the gold chip was selected as the region of interest (ROI) and an adjacent region without virions was selected as a reference region to remove common noises, such as drifting and light source noise. The image intensity of ROI and reference region were recorded by the SPRM 200 software in real time, and the differential value between the two regions was used to calculate the oscillation amplitude of the virion-oscillators. A homemade drug perfusion system was used to deliver ligands or buffer to the virion-oscillators. Binding curves were fit using the first-order kinetics model and calculated kinetics parameters[24,49,50].

**Pathogen assays on VirD-GPCR arrays**. Group B *Streptococcus* (GBS) K79 and *Streptococcus gordonii* (non-pathogenic control) were grown overnight in Todd-Hewitt (TH) broth (Difco Laboratories, Detroit, MI). Both strains were washed twice in PBS and determined the cell concentration by $OD_{600}$ $1 = 5 \times 10^8$. $2 \times 10^9$ bacteria were mixed with 5 nmol Cy3-NHS ester for 30 min. The free Cy3-NHS ester was quenched with 1 M Tris at pH 7.0 and removed by three centrifugations. To reduce nonspecific attachment, microbial were resuspended in 50% PBS, 1.5% BSA, 50% protein-free blocking buffer (Pierce), and 0.1 mg mL$^{-1}$ heparin sulfate (Sigma-Aldrich) for 30 min. The GPCR-VirD array was thawed and blocked in 50% PBS with 1.5% BSA, 50% protein-free blocking buffer, and deglycosylation cocktail (NEB) for 1 h with shaking. After a brief wash, $5 \times 10^8$ bacteria were added to the VirD-GPCR array and incubated for 1 h. The arrays were then subjected to three 5-min washes in PBS containing 0.01% tween, air-dried, scanned with GenePix 4000B. The resulting signal to noise ratios were acquired and analyzed with GenePix software.

**CYSLTR1 inhibition in vitro and in vivo**. Bacterial strains were grown overnight in Todd-Hewitt broth (Difco Laboratories, Detroit, MI) and added to HBMEC in a MOI of 100 at 37 °C in 5% $CO_2$ for 1.5 h[51]. HBMEC were subsequently washed and incubated with experimental media containing penicillin G (5 μg mL$^{-1}$) and gentamicin (100 μg mL$^{-1}$) for 1 h to kill extracellular bacteria. The HBMEC were washed, lysed, and the released intracellular bacteria were enumerated by plating on sheep blood agar plates. The invasion results were calculated as a percent of the initial inoculum and expressed as percent relative invasion compared to percent invasion in the presence of vehicle control (DMSO). Each set was run in triplicates.

Each BALB/C mouse received $1 \times 10^8$ CFU of GBS in 100 μL PBS via the tail vein injection. One hour later, mouse chest was cut open, and blood from the right ventricle was collected and plated for bacteria counts, which were expressed as CFU mL$^{-1}$ of blood. The mouse was then perfused with a mammalian Ringer's solution by transcardiac perfusion. The brains were removed, weighed, homogenized, and plated for bacterial counts, which were expressed as CFU g$^{-1}$. All procedures and handling techniques were approved by the Animal Care and Use Committee of the Johns Hopkins University.

**Statistical analysis**. For studies using TIRF microscopy to measure the ligand binding to the GPCR virions, at least 10 biologically independent samples per group were used. For the ligand binding to the cells, three independent samples per group were used. All the sample sizes were described in the associated figure legends. Statistical comparisons were carried out using two-tailed Student's *t*-test with a $P < 0.05$ as the threshold for significance. All data were presented as mean ± SD, where *n* was the number of subjects.

**Reporting summary**. Further information on research design is available in the Nature Research Reporting Summary linked to this article.

## Data availability

The datasets generated during and/or analyzed during the current study are available from the corresponding author upon reasonable request. The list of GPCR we cloned was attached in the Supplementary Table 2. All of the VirD constructs will be available to the research community free of charge and all of the VirD reagents will be distributed via CDI Labs upon reasonable requests. Source data underlying Fig. 1b; Fig. 2b, d, f, g; Fig. 3a, b, d; Fig. 5a, b, c; Supplementary Figures 1d, 2b, 2d, 2f, 4, 6b are provided as Source Data file.

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

## Acknowledgements

This work was supported in part by the NIH/NCI IMAT R33-CA186790-01A1 (to H.Z. and P.J.D.), NIH TCNP Roadmap (to H.Z.), NIH R01AG061852 (to H.Z.), NIH NS091102, AI84984, AI113273, and AI126176 (to K.S.K) and AI126176 (to K.S.K), NIH R01NS054791 (to X.D.), Johns Hopkins University Discovery Awards (to J.X.), Hamilton Innovation Award (to J. X.), Taiwanese Government Scholarship Award (to S.-C.W.), CDI Laboratories, and MOST Postdoctoral Fellowship—Taiwan (to G.-D.S.). We would like to thank the following colleagues for their assistance and support: David Leib (Dartmouth University) for the gift of KOS-37 BAC; David Johnson at Oregon Health and Science University for the generous gift of anti-gD antibody, DL6. We also thank Jen Tullman and Wade Gibson (Johns Hopkins University) for helpful discussions on virion purification and BAC recombineering method.

## Author contributions

G.-D.S., S.-C.W., G.M., S.L., A.P., D.P., B.H., D.G., P.R., P.J.D. and L.-C.W. performed experimental work. G.-D.S., H.Z., P.J.D., K.S.K., N.T., S.W., D.E., I.P., X.D., J.X., S.-C.W., G.M. and A.P. contributed to manuscript preparation. N.T., S.W., D.E., I.P., X.D. and J.X. contributed their expertise and supervision to the work. H.Z., P.J.D. and K.S.K. conceived the idea and supervised the entire project. H.Z. and G.-D.S. wrote the manuscript.

## Additional information

**Competing interests:** H.Z., D.E. and I.P. are cofounders and shareholders of CDI Laboratories Inc. P.R. is an employee of CDI Laboratories Inc. H.Z., G.-D.S. and P.J.D. are consultants to CDI Laboratories Inc. The remaining authors declare no competing interests.

