## [Peer Review File · Nature Communications]

Reviewers' comments:

Reviewer #1 (Remarks to the Author):

Syu et al. developed a high throughput screening method for human GPCRs based on their display on HSV virions. They showed that the GPCRs are correctly displayed by several methods and confirmed binding of known ligands (antibodies and others) thereby identifying unknown off-target effects by many of them. Furthermore, applying this novel technology, authors identified a novel receptor for GBS-host interaction that could have important impacts for future therapeutic targets. The paper is well written and all methods and results are explained in a detailed manner. The method is of great novelty and of huge interest in the field, and also discussed with respect to other GPCR screen technologies. All experiments are well designed and explained; results presented are clear and convincing. I recommend publication (after minor revision) of the VirD-GPCRs technology, as it should greatly foster research on GPCR, e.g. deorphanization of GPCRs, measuring off target effects of antibodies, GPCR ligands and many more. The topic is of great interest to researchers of various disciplines.

Minor points:

- HSV-1 expresses viral GPCRs which are also incorporated in the viral membrane – could these viral GPCRs affect assay performance?
- Table 1 should give an overview about the tested antibodies but following the uniprot (not Uniport ID) just gives information on the GPCR. The tested antibodies incl. providers should be listed
- Supplementary Fig 4: P2RY13 (the ligand that did not work) is not shown, however 8 ligand binding assays are shown that are significantly different from the K082 control. It seems that this result does not fit to what is written in the results section.
- Figure 5a shows 5 panels showing that GBS bind to the respective receptors. It would be more informative to show examples of receptors that do not bind GBS rather than comparing GBS with *S. gordonii* binding.

Reviewer #2 (Remarks to the Author):

The authors undertook an heroic amount of work in the present manuscript, expressing more than 300 non-olfactory GPCRs using high content virion display using HSV-1. This will no doubt be a tremendous resource allowing new opportunities for drug screening in a relevant cellular context without compromising the speed of such screens. They also provide an excellent assessment of the abysmal state of commercial antibodies generally available for tracking GPCRs. The experiments are well-performed and well-analyzed and they demonstrate how the library can be used with an excellent example. I only have a few comments that they could address going forward.

1) What criteria were used to select the commercially available antibodies used? I think this would be a useful comment they could add in their description of these studies and their results.

2) The NHS dyes used to label peptides- did they leave the ligands functional? Some demonstration of this beyond the primary binding assay should be provided- a signalling assay for example. How did the affinities compare with radiolabeled compounds if available? Some confirmation that the labeling itself was not responsible for the off-target binding effects would be useful.

Reviewer #3 (Remarks to the Author):

This manuscript by Syu et al describes an approach to create an array of non-olfactory G protein-coupled receptors in a folded state, which would enable simultaneous interrogation of large parts of this protein family important in human disease. Building on prior work from the Zhu lab, the main idea is to affix protein epitopes in high density. Here, the authors develop a method to express 315 GPCRs in the membrane of herpes simplex virus, which is then affixed to a glass slide. A few test uses are demonstrated: 1) specificity of commercially available antibodies, 2) specificity of peptide ligands, and 3) identification of new GPCR targets for pathogenic bacteria. In general, the work is innovative and provides a new approach for massively parallel interrogation of this important family. A few concerns limit enthusiasm for publication.

1) It is a bit unfortunate that the authors describe this new method, but no plans for resource sharing are provided. As the work is funded by the NIH, broad dissemination and success of this approach would be enabled by available repositories of the VirD virion library at Addgene, ATCC, or other resources.

2) A parallel approach that has gained substantial traction in the field is the use of viral-like particles, which obviate the need for generation of true viruses. The authors cite that VLPs lack reproducibility

are less stable. Both claims require more justification. Like VirD, VLPs can also be generated from alternative cell lines. A more extensive comparison to VLPs, including specific citations for limitations would be helpful.

3) Fig 2g - unclear what D1 antagonist means here - is this a fluorophore labeled compound. If so, which one? Please provide commercial product numbers for various reagents used off the shelf

4) It is now well established that many chemokines can bind their receptors with two binding modes; one that simply requires a non-folded amino terminus and another that requires a fully native receptor. Much of the data for functionality testing of GPCRs relies on chemokine interactions with chemokine receptors, which could largely derive from the first binding mode not reliant on a folded receptor.

5) The authors incorrectly state that the delta opioid receptor (OPRD1) is the target for dynorphin A. The primary target for dynorphin is the related kappa opioid receptor (OPRK1), which is not mentioned. It is anticipated that most opioid peptides will have cross-reactivity for other related opioid receptors. To understand how well the assay is working, the authors should provide the identities of the next 4 or 5 receptors in Fig 3a. In general, the authors picked two relatively selective ligands. Perhaps a better set of experiments would have examined known peptides that are promiscuous (RFamide peptides for example) vs. those that are specific within a family (opioid peptides).

6) Fig 4 - please list Kd what the errors here are (SEM, SD etc). For 4c, is there really no error?

Reviewer #1:

Syu et al. developed a high throughput screening method for human GPCRs based on their display on HSV virions. They showed that the GPCRs are correctly displayed by several methods and confirmed binding of known ligands (antibodies and others) thereby identifying unknown off-target effects by many of them. Furthermore, applying this novel technology, authors identified a novel receptor for GBS-host interaction that could have important impacts for future therapeutic targets. The paper is well written and all methods and results are explained in a detailed manner. The method is of great novelty and of huge interest in the field, and also discussed with respect to other GPCR screen technologies. All experiments are well designed and explained; results presented are clear and convincing. I recommend publication (after minor revision) of the VirD-GPCRs technology, as it should greatly foster research on GPCR, e.g. deorphanization of GPCRs, measuring off target effects of antibodies, GPCR ligands and many more. The topic is of great interest or researcher of various disciplines.

Minor points:

1) HSV-1 expresses viral GPCRs which are also incorporated in the viral membrane – could these viral GPCRs affect assay performance?

Response: In human herpes viruses, only KSHV and CMV are known to encode viral GPCRs (Arvanitakis et al., 1997 PMID: 9002520; van Cleef et al., 2006 PMID: 16406796). One reason that we chose HSV-1 to display human GPCRs was because it does not encode any GPCR genes and therefore, would not interfere with engineered human GPCRs.

2) Table 1 should give an overview about the tested antibodies but following the uniprot (not Uniport ID) just gives information on the GPCR. The tested antibodies incl. providers should be listed

Response: We agree. The providers and the datasheet statements are now added to Table 1 (page 21).

3) Supplementary Fig 4: P2RY13 (the ligand that did not work) is not shown, however 8 ligand binding assays are shown that are significantly different from the K082 control. It seems that this result does not fit to what is written in the results section.

Response: Thank you for pointing out this confusing description. We have revised the sentence to “Except P2RY13, all the nine VirD-GPCRs that were not recognized by the commercial mAbs showed significantly higher binding signals to their canonical ligands than the K082 control, indicating that they were functional and folded correctly (Fig. 2g; Supplementary Fig. 4).” (page 7).

4) Figure 5a shows 5 panels showing that GBS bind to the respective receptors. It would be more informative to show examples of receptors that do not bind GBS rather than comparing GBS with *S. gordonii* binding.

Response: Two GPCRs, CNR1 and CASR, that did not bind GBS are now added in Fig. 5a as negative examples (page 26).

Reviewer #2 (Remarks to the Author):

The authors undertook an heroic amount of work in the present manuscript, expressing more than 300 non-olfactory GPCRs using high content virion display using HSV-1. This will no doubt be a tremendous resource allowing new opportunities for drug screening in a relevant cellular context without compromising the speed of such screens. They also provide an excellent assessment of the abysmal state of commercial antibodies generally available for tracking GPCRs. The experiments are well-performed and well-analyzed and they demonstrate how the library can be used with an excellent example. I only have a few comments that they could address going forward.

1) What criteria were used to select the commercially available antibodies used? I think this would be a useful comment they could add in their description of these studies and their results.

Response: The criteria used to select the 20 commercial antibodies were the following: i) They have to be monoclonal antibodies; ii) They should recognize the ectodomains of the human GPCRs; and iii) They have to be flow-positive when performed against intact cells, based on the datasheet provided by the commercial source. We have now added these descriptions to the revised manuscript (page 5). In addition, we now listed the providers' information in Table 1.

2) The NHS dyes used to label peptides- did they leave the ligands functional? Some demonstration of this beyond the primary binding assay should be provided- a signalling assay for example. How did the affinities compare with radiolabeled compounds if available? Some confirmation that the labeling itself was not responsible for the off-target binding effects would be useful.

Response: We employed calcium-imaging methods to compare the differences of activation of GPCRs between dye-labeled and -unlabeled ligands in transfected cells and no significant differences were observed. These new data are now described on page 7 and shown in Supplementary figure 5. The K_i values of SRIF-14 and SSTR2 are in the low nM range (Taylor et al., 1994 PMID: 7854974), which are similar to our measurement.

Reviewer #3 (Remarks to the Author):

This manuscript by Syu et al describes an approach to create an array of non-olfactory G protein-coupled receptors in a folded state, which would enable simultaneous interrogation of large parts of this protein family important in human disease. Building on prior work from the Zhu lab, the main idea is to affix protein epitopes in high density. Here, the authors develop a method to express 315 GPCRs in the membrane of herpes simplex virus, which is then affixed to a glass slide. A few test uses are demonstrated: 1) specificity of commercially available antibodies, 2) specificity of peptide ligands, labeled and 3) identification of new GPCR targets for pathogenic bacteria. In general, the work is innovative and provides a new approach for massively parallel interrogation of this important family. A few concerns limit enthusiasm for publication.

1) It is a bit unfortunate that the authors describe this new method, but no plans for resource sharing are provided. As the work is funded by the NIH, broad dissemination and success of this approach would be enabled by available repositories of the VirD virion library at Addgene, ATCC, or other resources.

Response: Thank you for pointing this out. We will share this resource with the research community free of charge after this manuscript is accepted for publication. In addition, the entire VirD-array technology has been licensed to CDI Laboratories, Inc., and all the relevant constructs will be commercially available at CDI after this work is published.

2) A parallel approach that has gained substantial traction in the field is the use of viral-like particles, which obviate the need for generation of true viruses. The authors cite that VLPs lack reproducibility are less stable. Both claims require more justification. Like VirD, VLPs can also be generated from alternative cell lines. A more extensive comparison to VLPs, including specific citations for limitations would be helpful.

Response: We have added more references to justify our discussion with regard to the comparison between VLP and VirD in the Discussion section (page 10-11).

3) Fig 2g - unclear what D1 antagonist means here - is this a fluorophore labeled compound. If so, which one? Please provide commercial product numbers for various reagents used off the shelf

Response: Yes, D1 antagonist is a commercial name of the fluorophore-labeled compound, SKF83566 (The provider and catalog number was added in the method section on page 17).

4) It is now well established that many chemokines can bind their receptors with two binding modes; one that simply requires a non-folded amino terminus and another that requires a fully native receptor. Much of the data for functionality testing of GPCRs relies on chemokine interactions with chemokine receptors, which could largely derive from the first binding mode not reliant on a folded receptor.

Response: Thank you for pointing this out. Although we could not distinguish which binding mode contributed more to the interactions with the chemokines tested in our assays, we are confident that most of the VirD-GPCRs should fold correctly because we demonstrated here and in a previous study (Ma et al., 2018 PMID: 30114365) that many small molecule ligands could be captured by their canonical GPCRs on virions.

5) The authors incorrectly state that the delta opioid receptor (OPRD1) is the target for dynorphin A. The primary target for dynorphin is the related kappa opioid receptor (OPRK1), which is not mentioned. It is anticipated that most opioid peptides will have cross-reactivity for other related opioid receptors. To understand how well the assay is working, the authors should provide the identities of the next 4 or 5 receptors in Fig 3a. In general, the authors picked two relatively selective ligands. Perhaps a better set of experiments would have examined known peptides that are promiscuous (RFamide peptides for example) vs. those that are specific within a family (opioid

peptides).

Response: We agree that OPRK1 is the canonical receptor for dynorphin A; however, OPRD1 was also reported to interact with dynorphin A (Zhang et al., 1998; PMID: 9655852). To demonstrate that dynorphin A can activate OPRD1, we performed a calcium imaging experiment using OPRD1-infected cells and showed that dynorphin A could indeed induce Ca^{2+} influx, although to a lesser extent than OPRK1-infected cells (Supplemental figure 5). To better understand why OPRK1 did not show detectable binding signals to dynorphin A on the VirD-GPCR arrays, we measured its expression level in purified viruses using anti-V5 antibodies and found that it was present at a very low level (Supplemental figure 6a). To rule out the possibility that the kappa receptor was not correctly folded in the envelope of the virions, we employed a more sensitive TIRF microscopy method and demonstrated that VirD-OPRK1 showed significantly higher binding signals to Cy3-labeled dynorphin A than the K082 control virions (Supplementary figure 6b). Taken together, this new evidence confirmed that dynorphin A could activate both kappa and delta receptors in cells and that virion-displayed kappa receptor could bind to dynorphin A. We have now added these new data to the manuscript on page 7 and in Supplemental figures 5&6. We have also added the identities of the next 2 receptors in Figure 3a.

6) Fig 4 - please list K_d what the errors here are (SEM, SD etc). For 4c, is there really no error?

Response: We have added the errors to Figure 4c and Supplemental Table 1.

REVIEWERS' COMMENTS:

Reviewer #1 (Remarks to the Author):

Authors have addressed all issues in their revised manuscript.

Reviewer #2 (Remarks to the Author):

Thanks for responding to my questions.

Reviewer #3 (Remarks to the Author):

The authors have reasonably answered my technical concerns.

I am still bothered by the lack of clear resource availability in the "Data availability" section. While the authors suggest they will make these resources available free of charge after publication, this is not reflected in the Data availability section. There is a clear conflict of interest (which is appropriately disclosed) to making such resources freely available given that the senior author is a founder of the company selling the reagents.

Point by point response to the reviewers

Reviewer #3 (Remarks to the Author):

The authors have reasonably answered my technical concerns. I am still bothered by the lack of clear resource availability in the "Data availability" section. While the authors suggest they will make these resources available free of charge after publication, this is not reflected in the Data availability section. There is a clear conflict of interest (which is appropriately disclosed) to making such resources freely available given that the senior author is a founder of the company selling the reagents.

Response: Thanks for the suggestion. We add "All of the VirD constructs will be available to the research community free of charge" to the Data Availability section.